# Liquid-Infused Porous Film Self-Assembly for Superior Light-Transmitting and Anti-Adhesion

**DOI:** 10.3390/mi13040540

**Published:** 2022-03-30

**Authors:** Yang Liu, Xiaoyang Zhan, Yan Wang, Guang Liu, Deyuan Zhang, Liwen Zhang, Huawei Chen

**Affiliations:** School of Mechanical Engineering and Automation, Beihang University, Beijing 100191, China; liuyang168@buaa.edu.cn (Y.L.); zy2007429@buaa.edu.cn (X.Z.); wangyan1119@buaa.edu.cn (Y.W.); liuguang0701@buaa.edu.cn (G.L.); zhangdy@buaa.edu.cn (D.Z.)

**Keywords:** self-assembly, well-ordered porous film, liquid-infused surface, improved durability, light-transmitting, anti-adhesion

## Abstract

Liquid-Infused Surfaces (LISs), particularly known for their liquid-repelling feature, have demonstrated plenty of applications in the medical, marine, and energy fields. To improve the durability and transparency highly demanded on glass-based vision devices such as an endoscope, this study proposed a novel self-assembly method to fabricate well-ordered porous Poly-Styrene (PS)/Styrene–Butadiene–Styrene (SBS) films by simply dripping the PS/SBS dichloromethane solutions onto the glass before spinning. The effects of the solutions’ concentrations and spin speeds on the porous structure were experimentally investigated. The results showed that a certain mass ratio of PS/SBS can make the structure of the ordered porous film more regular and denser under the optimal solution concentration and spin-coating speed. Superior transparency and durability were also realized by dripping silicone oil on the porous film to build a liquid-infused surface. Applications of the as-prepared surface on devices like endoscopes, viewfinders, and goggles have been explored respectively.

## 1. Introduction

Camera-guided instruments are extensively used in a variety of fields such as oil and marine explorations, sanitary inspections, robotics, optical sensors, and medicines. However, their operation is severely compromised in highly contaminated environments where oil, sewage, marine fouling, or body fluids permanently spoil the view. The cleansing methods developed for these applications are not sufficient to maintain the light transmittance constancy as their continued performance is time-consuming, expensive, and often ineffective. The primary means used to maintain clean lenses usually depends on mechanical wiping, in which case contours or curvature are particularly difficult to clean. Mechanical wiping can also damage and wear the lenses over time, and in applications demanding attention to detail, damages and wears can distort the view and reduce the durability. Another approach to lens cleansing is to equip a camera with additional channels through which irrigation or spraying can be applied, which significantly increases the size of the instrument and limits the range of its application.

As a promising method, a novel antifouling, transparent coating, or thin-film material that can be applied to the surface of lenses may obviate contamination-induced vision loss, render cleaning procedures unnecessary, and allow miniaturization of the instrument used in previously inaccessible environments or confined spaces. Perhaps more notably and more importantly than any other industrial application, this solution is embodied in medical procedures, such as endoscopy, viewfinders, and so on [1,2,3,4].

Surface modification has been usually adopted to adjust the material and chemical properties of the lens. For example, Yu et al. [5] formed microstructures on glass by using a plasma etching method to generate a superhydrophilic surface which mimics the self-cleaning of lotus leaf. Although superhydrophilic surfaces were widely exploited, Liquid Infused Surfaces (LISs) have attracted much attention from scientists recently because of their special surface wetting properties [6,7]. Compared to other strategies, LISs have advantages due to their availability, relatively low cost, high processing speed, and wide variety of patterning effects. For instance, Sunny et al. [4] reported an anti-fouling material fabricated via the layer-by-layer (LbL) deposition of charged particles to form hierarchical micro porous structures with infiltrated lubricants. The emergence of LISs raises possibilities for new applications and optimizes material properties in multiple fields. LISs have shown good application prospects in anti-corrosion [8,9,10], anti-icing [11,12], biomedical applications [4,13,14], self-healing [15,16,17], droplet manipulation [18,19,20]. etc. However, LISs also have some aspects to be improved. For example, the stability of the lubricating effect of the lubricating fluid needs to be strengthened. Moreover, LISs can be used as an anti-adhesion method for optical lenses, but their durability problem also needs to be solved before [21].

The retention of liquids by ordered structured films can be adopted as a way to improve durability and has been potentially applied in various fields of optoelectronic devices [22], membrane filtration [23], tissue engineering [24], superhydrophobic surfaces [25], catalysis [26,27], etc., which is expected to solve the above-mentioned contamination problems of the lens surface. However, there are also some shortcomings, for example, the anti-adhesive and anti-fouling effects have not yet met the technical requirements, and the coating has poor reliability showing a tendency to peel off; the anti-adhesive ability of the micro-nano structure of the surface is limited, and the surface itself is not strong enough and may easily be damaged [28,29]. A variety of top-down methods, such as standard microelectronic fabrication processes (e.g., plasma etching, and photolithography) and soft lithography have been applied to fabricate well-ordered porous films [5,30]. However, these methods suffer a certain number of disadvantages, such as being high-cost, time-consuming, and complicated. Compared with the abovementioned top-down approaches, self-assembly ones, including micro-phase separation [31,32], sol–gel [33,34,35], templating [36] and nanoparticle assembly [37], are proven to be simpler and more effective.

Herein, an effective strategy to synthesize well-ordered porous films has been developed by mixing PS and SBS at a certain ratio, dissolving it in dichloromethane, and then spin-coating the solution on a glass. The well-ordered porous-structured film can be obtained by systematic self-assembly. The surface structural characteristics of the films were individually generated by varying the concentration of PS/SBS solution as well as the speed and time of spin-coating. This method enables the production of porous films directly on either flat or curved surfaces with excellent light transmission, durability, and anti-adhesive properties after the injection of lubricant, which can be applied to the production of lenses for camera-guided instruments.

## 2. Materials and Methods

PS (Mw = 150,000) and SBS (KTR-101, 80,000 ≤ Mw ≤ 120,000, 30 wt % PS) were purchased from Sigma-Aldrich (Shanghai, China) Trading Company Limited (Shanghai, China) and Kumho Petro chemical (Seoul, Korea), respectively. Dichloromethane, acetone, n-hexane (A.R), and Ethanol (G. R.) were purchased from Beijing Chemical Works (Beijing, China). The perfluoropolyether lubricant (DuPont Krytox GPL100, surface tension 17 mNm^−1^), viscosity 0.124 cm^2^ s^−1^ at 20 °C) was purchased from DuPont Corporation (Wilmington, DE, USA). All chemicals were applied as received. The water used throughout the experiments was deionized. The blood used in the experiments came from laboratory mice. The experimental devices included a 12A Benchtop Spin Coater (Shandong Guan Brand); electron microscope (SEM, SU8000, Hitachi, Tokyo, Japan); an angle measuring system (SL200B, Solon, Shanghai, China); a spectrophotometer (UV-3600, Shimadzu, Japan); a profilometer (KLA Tencor D-600, KLA Inc., Milpitas, CA, USA); a test endoscope (AGU-100, Hangzhou Mirror Tour Technology Co., Ltd., HangZhou, China); and a syringe (0.45*16RWSB, Jiangsu Great Wall Medical Equipment Co., Yangzhou, China)

For the production of porous structures with PS and SBS, the effect of PS-to-SBS mass ratio on the pore structure has already been reported in our previous research [38], so the above factors need not be reconsidered in this experiment. Based on the results of previous experiments, the optimal ones were selected to apply in this experiment, as shown in Figure 1. PS and SBS were mixed in a beaker at the ratio of 3:1 (Figure 1a), and a certain amount (20–30 mL) of methylene chloride was added in (Figure 1b). Then, it was sealed and kept in the dark at room temperature for 48 h so that PS and SBS could be completely dissolved, and the volatilization of methylene chloride would be reduced (Figure 1c). Circular curved glass substrates used for deposition were ultrasonically cleaned in hexane, acetone, and alcohol for 15 min and then dried at 50 °C for 30 min under a vacuum. The ambient temperature was controlled at 18 to 20 °C. A volume of 0.2 mL of the prepared solution was added dropwise onto a circular curved glass substrate (30 mm in diameter), before being uniformly spin-coated by a bench-top spin coater to obtain a uniformly and orderly porous film structure during the later drying process of the solution (Figure 1d). After coating, the glass substrate with solution was dried at room temperature for 3–5 min to obtain a uniformly and orderly porous film structure (Figure 1e). Subsequently, 20 μL cm^−2^ silicone oil was added onto the glass substrate until spread over the entire surface. Finally, the substrates were positioned vertically for 3 h to drain off the excess silicone oil (Figure 1f–g). Contact angles (CAs) and sliding angles (SAs) were measured by use of the droplet volume of 5 μL. The transmittance of the samples over the entire visible spectrum, 400–800 nm, was recorded using a spectrophotometer. The film thickness was measured by a profilometer.

## 3. Results

### 3.1. Formation Mechanism

The formation of porous films could be explained as follows: As shown in Figure 2, a certain concentration of PS/SBS solution can remain stabilized on the glass to some extent without spreading instantaneously (Figure 2a). After the solution finally spreads over the glass, the PS blocks start to aggregate as the dichloromethane evaporates. With further evaporation of the dichloromethane (Figure 2b), the PS interactions lead to the formation of holes (Figure 2c), as explained by Mattehew’s group [39]. During the self-assembly, three main inter-molecular forces exist between the particles of the solution: van der Waals forces, hydrophobic forces, and hydrogen bonding forces [40]. The repulsive forces between the PS blocks and the PS phases in SBS are the dominant factors in the self-assembly process. To maintain the equilibrium, the structure compromises between complete order and complete disorder. The most stable structure has been the isohexagonal one, a state that is never reachable due to entropy. In the present experiments, the porous structure was finally formed (Figure 2d). The side-view SEM photo of the porous structure is shown in Appendix A. For SBS, the hard phase and soft phase of the PS segment were both included, while the PB segment mainly assisted the formation of ordered films.

In this self-assembly, as for SBS, which contains the PS hard phase and the PB soft phase [38], the major internal driving force comes from the repulsive force between PS (PS and the PS phases in SBS). Compared to the pure PS, the addition of a small amount of SBS causes an imbalance in the system, and the small content of the introduced PB phase is not enough to counteract the repulsive force between PS, thus leading to the disorder in the structure. When the SBS rises to such an amount that the PB phase is sufficient to offset the repulsive force between PSs, the structure becomes more orderly. The purpose of the spin coating is to adjust the film thickness for better light transmission performance; it also helps to accelerate the volatilization of methylene chloride, which in turn quickly forms a porous film structure.

### 3.2. Influence of Concentration

The solution concentration can affect the surface morphology of the film to a great extent [41]. To investigate the effect on the film pore structure and its surface morphology, the solution concentration was varied from 20 to 100 g/L at a fixed PS/SBS mass ratio of 3:1. As shown in Figure 3a, the pores on the film were distributed in a disorderly manner, and the pore size only varied at solution concentrations greater than 60 g/L. At concentrations below 40 g/L, the pore distribution on the membrane became gradually sparse.

The degree of cross-linking between PS blocks varies with the concentration changes of the solution. As mentioned above, a certain amount of SBS contributes to the formation of ordered structures in this hybrid process, but if the SBS exceeds this amount, the opposite effect is observed. At greater solution concentrations, the degree of cross-linking between PS is also greater, leading to a reduction in the kinetics of the polymer chains. With the further evaporation of the solvent, less energy is not conducive to polymer aggregation [42] so that the solution generally exhibits irregularities at higher concentrations. At lower concentrations, there is more energy between PSs propelling them to interact with each other and form ordered structures. However, at excessively low concentrations, the small amount of PB phase is not enough to counteract the repulsive forces between PS, thus again leading to a disordered structure, which in turn generates fewer porous structures with the irregular distribution.

### 3.3. Influence of Spin Coating Speed

Spin coating speed directly determines the film’s flatness and uniformity. If the spin coating is not applied or the applied speed was too low, film flatness and uniformity is not guaranteed. As a consequence, the solution does not spread evenly on the glass, forming a very irregular pore structure, as shown in the first two pictures in Figure 3b, and the average pore size and pore pitch both fluctuate widely, as shown in Appendix A. Moreover, the thickness of the film is large and the light transmission is poor, as shown in Figure 4a. When the thickness of the film increases, more light is absorbed, and therefore less light is transmitted through, and light transmission decreases [43]. When the spin-coating speed is too fast, although the light transmission is ensured, the solution on the glass is thrown out and thus cannot form a regular and continuous hole structure, as shown in the last two figures in Figure 3b, which process at spin coating speeds of 500 r/min and 700 r/min, respectively. At a suitable spin-coating speed (300 r/min), the effect is just sufficient for both the polymerization and distribution, which in turn forms a regular porous structure. As shown in Appendix A, under the parameters of optimum concentration of 40 g/L, spin-coating speed of 300 r/min and spin-coating time of 6 s, the pore size and pore pitch values presented the least fluctuations. Combined with the above analysis, the anti-adhesive performance is the best under this case, so the following performance application tests are conducted with these parameters.

### 3.4. Light Transmission Performance

For camera-guided devices, the visibility of the lens is essential to maintain their effective work. If the well-ordered porous film surface has excellent light transmission ability, it can be applied to surfaces such as endoscopes. Compared to well-ordered porous films formed by self-assembly at the gas–liquid interface, the film thickness cannot be tuned, and thus, the resulting film presents the satisfying performance of the light transmission. As mentioned above, this processing method can adjust the film thickness and thus affect the light transmission by changing the spin coating speed. The transparencies of the flat glass substrate, dry porous film, and super wet slip porous film (dealt with silicone oil) were compared under the parameters of concentration of 40 g/L, a spin-coating speed of 300 r/min, and a spin-coating time of 6 s which were determined as the optimum in the previous section, as shown in Figure 4b. It can be noted that spin coating improves the porous film’s light transmission, and the transparency of wet porous film is better than that of the dry porous film. It is roughly 89%, basically the same as that of the glass substrate, proving the transparency of wet porous film surface has been significantly improved. In fact, the addition of silicone oil reduces the scattering of light in the small pore structure and exerts the effect of enhancing transparency. Figure 4c shows a comparison of the transparencies of the dry porous film (right) and the wet porous film surface (left) on flat glass. The wet porous film surface clearly shows better transparency, which indicates that the excellent optical properties are based on the super slippery surface of the porous film. As to the curved convex mirror shown in Figure 4d, porous film structures can also be applied by spin-coating directly on its curved surface, and its light transmission effect has also been compared (Figure 4d) and found to be consistent with that of the flat surface.

### 3.5. Anti-Adhesive Properties

The wettabilities of seven different droplets were compared on the glass substrates of porous films with and without silicone oil. A volume of 20 μL cm^−2^ silicone oil was added onto the glass substrates until it was spread over the entire surface. Finally, the substrates were positioned vertically for 3 h to drain off the excess silicone oil. As shown in Figure 5a and Appendix A, the wet-slip porous film consistently showed a reduced contact angle compared to the dry-slip porous film and a low slip angle ≤ 5°. These results indicate that the silicone oil likely forms a uniform film on the surface and thus significantly reduces the effect of the structure on the wettability. The super slippery properties of the surface of the well-ordered porous film were proved to be widely applicable.

By examining the adhesive performances of blood droplets on the surface of the self-assembled porous film plus silicone oil and on the surface of the bare glass, the excellent anti-adhesive properties of the well-ordered porous film structure were confirmed. As shown in the consecutive images in Figure 5b,c, blood droplets (10 μL) were dropped onto each surface at an angle of 5° to observe its specific surface adhesion during sliding. It can be observed that the blood drops slid on the bare glass for a certain period (3 s) and then became stationary (Figure 5b), thus leaving blood traces on the surface. In contrast, blood drops could slide on the porous film with silicone oil for less than 5 s without leaving any stain or residue (Figure 5c), demonstrating the excellent anti-adhesive properties of the ordered porous film.

The stability and reusability of ordered porous film surfaces directly determine whether they can be functional in a wide range of applications. Therefore, their relevant properties were explored as shown in Figure 6a. The durability of the anti-adhesive properties of the porous film was investigated by comparing the sliding velocities of blood droplets on the surface between the wet porous film and the bare glass with silicone oil. As shown in Figure 6a, the sliding speed of blood droplets on the porous film surface was significantly faster than that on the bare glass surface, which verified the above analysis that the micro-pores on the surface were beneficial to improving the sliding durability. Although the bare glass surface coated with silicone oil could also maintain good lubricity in the previous experiments, the lubricant was consumed by the droplets as the number of experiments increased. Therefore, after cycle tests, the anti-adhesive property of bare glass was significantly weakened because the blood droplet failed to slide down the bevel, as shown in Figure 6b. It can be deduced that the structure of the porous film can hold the lubricating fluid better than that of ordinary bare glass, manifesting the improved lubrication durability combined with the anti-adhesive performance.

### 3.6. Comprehensive Performance

The application test for the comprehensive performance of the well-ordered porous film was performed by simulating the light transmission and anti-adhesive performances in the face of blood spray during endoscopic surgery. Figure 6c shows the endoscope adopted for the test. Before the start, the endoscope piece was disassembled, and the porous film was spin-coated on its curved surface. Then it was installed back, and silicone oil was dripped on the lens surface. The adsorption test of sprayed blood with a syringe at the endoscope is shown in Figure 6d, with the graph in the rightmost column indicating the final visible area of the endoscopic lens. The first row of images in Figure 6d shows blood droplets being adsorbed onto the exposed endoscope lens and then dispersed due to surface hydrophilicity, resulting in immediate and complete loss of vision. In contrast, the following images show the excellent anti-adhesive properties of the ordered porous film structure, which leads the adsorbed blood droplets to slip off quickly. It in turn facilitates the rapid recovery of visibility.

## 4. Discussion

A new method to fabricate well-ordered PS/SBS porous films by spin coating on a glass substrate with PS/SBS solution applied dropwise has been developed. The optimum concentration of PS/SBS solution is 40 g/L, the optimum spin-coating speed is 300 r/min, and the optimum spin-coating time is 6 s. The film presents good light transmission performance after lubricant treatment at a rate similar to that of bare glass, while the anti-adhesive property is also greatly improved. When the glass surface is fixed at a 5° tilt angle, the contaminant under test (i.e., blood) can be easily removed from the film surface. In an experiment, a porous film was formed by spin coating on the endoscope lens for examination, and its effectiveness in preventing blood adhesion was confirmed; thus, it was verified as maintaining clear visibility of the endoscopic view. This approach allows the production of porous film structures directly on free-form surfaces, and its simplicity can provide an ideal way to design camera-guided devices, such as lenses for medical and engineering endoscopes.

## Figures and Tables

**Figure 1 micromachines-13-00540-f001:**
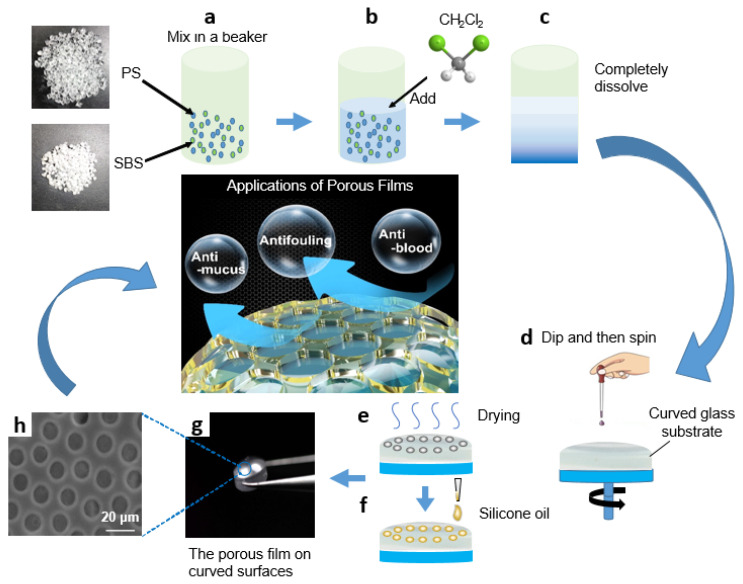
Schematic diagram of PS/SBS porous film formation: (**a**) PS and SBS mix in a beaker at the ratio of 3:1; (**b**) CH_2_Cl_2_ is added; (**c**) PS and SBS are completely dissolved; (**d**) the prepared solution is dripped onto a circular curved glass substrate and spun; (**e**) the glass substrate with solution is dried; (**f**) silicone oil is added onto the glass substrate; (**g**) the orderly porous film is obtained on curved surface; (**h**) the SEM photo of porous film.

**Figure 2 micromachines-13-00540-f002:**
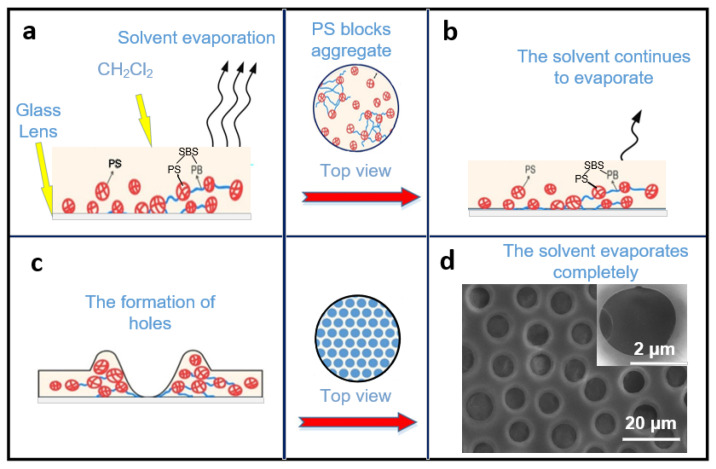
Process of the porous film formation mechanism: (**a**) PS/SBS solution remains stabilized on the glass and solvent evaporate. (**b**) The dichloromethane continues to evaporate. (**c**) The PS interactions lead to the formation of holes. (**d**) The porous structure is formed after solvent evaporate completely.

**Figure 3 micromachines-13-00540-f003:**
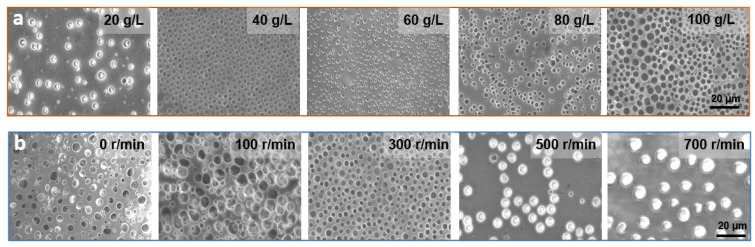
(**a**) SEM images of PS/SBS porous films formed at different concentrations. (**b**) SEM images of PS/SBS porous films formed at different spin coating speeds. Other conditions: the fixed weight ratio of PS/SBS:3:1. Spinning speed: 300 r/min, time: 6 s.

**Figure 4 micromachines-13-00540-f004:**
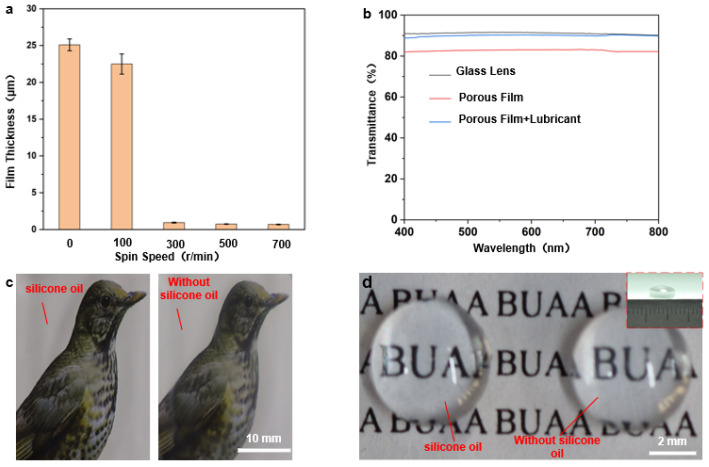
(**a**) Graph of spin-coating speed in relation to film thickness (the picture above the graph shows the comparison between the light transmission rates of the corresponding films); (**b**–**d**) Optical transparency analysis of porous films: (**b**) transmission rates of dry and wet porous films at different visible spectra; (**c**) optical photographs of porous films on flat glass before and after the addition of silicone oil (the right side is without silicone oil); (**d**) optical photographs of porous films on curved convex mirrors (diameter 6 mm ) before and after the addition of lubricating oil (without silicone oil on the right).

**Figure 5 micromachines-13-00540-f005:**
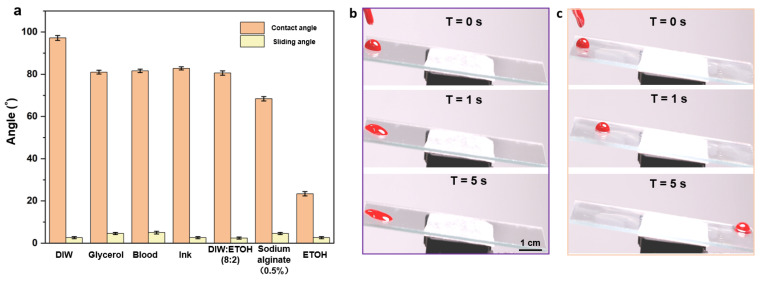
(**a**) Contact and slip angles of different droplets on the surface of a wet porous membrane; (**b**,**c**) sequential photographs of blood droplets (volume10 μL) on bare glass (**b**) and the glass with dealt porous film (**c**) at a tilt angle of 5°.

**Figure 6 micromachines-13-00540-f006:**
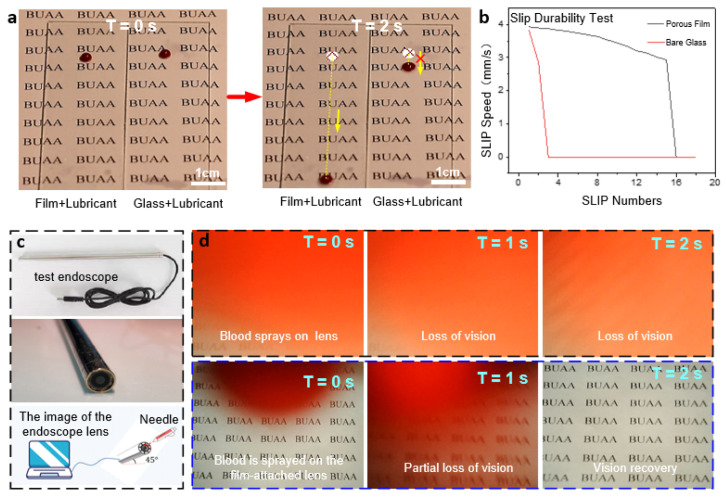
(**a**) Experimental graphs comparing the anti-adhesive performances of the porous film and ordinary glass. (**b**) Comparison curves of slip velocity between the two cases. (**c**) Test endoscope. (**d**) Endoscopic blood spray’s anti-adhesive test graphs (continuous images of the visual fields of the untreated lens (**top**) and lens with porous film (**bottom**)).

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
