# Peer review of "Liquid-Infused Porous Film Self-Assembly for Superior Light-Transmitting and Anti-Adhesion"

_micromachines, 2022, doi:10.3390/mi13040540_

Round 1

Reviewer 1 Report

Dear Sir, The comments about the following manuscript are appended below. Manuscript ID: micromachines-1646713 ArticleTitle: “Spin-coating Self-assembly of Liquid-Infused Porous Film with Superior Light-transmitting and Anti-adhesion” In this article, the authors reported a novel self-assembly method to fabricate well-ordered porous Poly-Styrene (PS)/Styrene–Butadiene–Styrene (SBS) films by simply dipping the PS/SBS dichloromethane solutions onto the glass before spinning. The effects of the solutions’ concentrations and spin speeds on the porous structure were also experimentally investigated. The authors are requested to respond the following queries. 1)The title should be properly changed. The present title is not meaningful. 2)The authors fabricated a polymer film consisting PS and SBS followed by coating a silicone polymer. The methodology given here not explain anything about the coating of silicone oil. 3)In so many places, the authors mentioned the formation of a porous film as a self assembled system. Shall we say, this as a self-assembly? 4)In section 3.1, page no.4, line no.124, the authors stated that, “…. and the small con- 124 5)tent of the introduced PB phase is not enough….”, what is PB phase? 6)In section 3.4, line no 156,157, the authors stated the sentence “In fact, the addition of lubricant reduces the scattering of light in the small pore structure and exerts the effect of enhancing transparency.” What do you mean by -addition of lubricant? 7)In section 3.5, the talked about “…the surface of porous films spiked with silicone oil. “ What is it means? How much silicone oil is used? Whether it is firmly attached? 8)In section 3.6, the authors coated the same constituents over endoscope head? How it was done? What is the shape? Normally, spin coating is applied over a flat specimen. 9)The non-adherent nature of the coating is not properly explained scientifically. Throughout the manuscript, the authors explained only the findings. Just applying silicone oil over a polymer film shows super hydrophobic nature is a well known fact, not a novel thing. So, the authors should explain in what way it is advantageous than the literature methods. 10)Whether the silicone oil coated film is durable? After including all these points in the revised manuscript, the paper can be accepted for publication. Reviewer.

Author Response

  1. Point 1: The title should be properly changed. The present title is not meaningful.

Response 1: Thanks for your comment. The title was adjusted and just focused on porous film self-assembly, as “Liquid-Infused Porous Film Self-assembly for Superior Light-transmitting and Anti-adhesion”.

  1. Point 2: The authors fabricated a polymer film consisting PS and SBS followed by coating a silicone polymer. The methodology given here not explain anything about the coating of silicone oil.

Response 2: Thanks for your kind comment. The relevant descriptions have bee In this self-assembly n modified as shown in Page 2, line 98 “20μL cm2 silicone oil was added onto the glass substrate until spread over the entire surface. Finally, the substrates were positioned vertically for 3 h to drain off the excess silicone oil”.

  1. Point 3: In so many places, the authors mentioned the formation of a porous film as a self assembled system. Shall we say, this as a self-assembly?

Response 3: Thanks for your kind comment. The relevant descriptions have been modified as shown in corresponding position of the paper. “During the self-assembly”、 “In this self-assembly”.

  1. Point 4: In section 3.1, page no.4, line no.124, the authors stated that, “…. and the small content of the introduced PB phase is not enough….”, what is PB phase?

Response 4: Thanks for your kind comment. The relevant descriptions have been modified as shown in Page 4, line 122. “In this self-assembly, as for SBS, which contains the PS hard phase and the PB soft phase [38]”.

  1. Point 5 In section 3.4, line no 156,157, the authors stated the sentence “In fact, the addition of lubricant reduces the scattering of light in the small pore structure and exerts the effect of enhancing transparency.” What do you mean by addition of lubricant?

Response 5: Thanks for your kind comment. The addition of lubricant is silicone oil. The relevant descriptions have been modified as shown in corresponding position of the paper.

  1. Point 6: In section 3.5, the talked about “…the surface of porous films spiked with silicone oil. “ What is it means? How much silicone oil is used? Whether it is firmly attached?

Response 6: Thanks for your kind comment. The relevant descriptions have been modified as shown in Page 6, line 202. “20μL cm−2 silicone oil was added onto the glass substrates until spread over the entire surface. Finally, the substrates were positioned vertically for 3 h to drain off the excess silicone oil. ”Later durability tests had proven to be able to adhere firmly as shown in Fig.6(b).

  1. Point 7: In section 3.6, the authors coated the same constituents over endoscope head? How it was done? What is the shape? Normally, spin coating is applied over a flat specimen.

Response 7: Thanks for your good comment. The relevant descriptions have been modified as shown in Page 7, line 244.“Before the start, the endoscope piece was disassembled, and the porous film was spin-coated on its curved surface. Then it was installed back and silicone oil was dripped on the lens surface”

  1. Point 8: The non-adherent nature of the coating is not properly explained scientifically. Throughout the manuscript, the authors explained only the findings. Just applying silicone oil over a polymer film shows super hydrophobic nature is a well known fact, not a novel thing. So, the authors should explain in what way it is advantageous than the literature methods.

Response 8: Thanks for your good comment. Although applying silicone oil over a polymer film shows super hydrophobic nature is a well known fact, our proposed preparation process is very simple, which allows the production of porous film structures directly on free-form surfaces. It is not the focus of other scholars.

  1. Point 9: Whether the silicone oil coated film is durable?

Response 9: Thanks for your good comment. The durability tests had proven the silicone oil coated film is durable as shown in Fig.6(b).

  1. Point 10: For the introduction part, the logic between paragraphs is not clearly shown. And at the end of the second paragraph of the introduction, there is a problem with the placement of punctuation and citations.

Response 10: Thanks for your comment. The relevant descriptions have been modified as shown in Page 1, line 42. “Surface modification has been usually adopted to adjust the material and chemical properties of the lens. For example, Yu et al. [5] formed microstructures on glass by using a plasma etching method to generate a superhydrophilic surface which mimics the self-cleaning of lotus leaf. Although superhydrophilic surfaces were widely exploited, Liquid Infused Surfaces(LISs) have attracted much attention from scientists recently because of their special surface wetting properties [6,7]. Comparing to other strategies, LISs have advantages due to its availability, relatively low cost, high processing speed, and wide variety of patterning effects. For instance, Sunny et al. [4] reported an anti-fouling material fabricated via the layer-by-layer (LbL) deposition of charged particles to form hierarchical micro porous structures with infiltrated lubricants.”

  1. Point 11: Introduction should also mention some other strategies beside liquid-infused surfaces (LISs) and describe the advantages of LISs over others. Besides, the introduction also needs to include at least one prototypical work of LISs with more details.

Response 11: Thanks for your kind comment. The relevant descriptions have been modified as shown in Page 1, line 42. “Surface modification has been usually adopted to adjust the material and chemical properties of the lens. For example, Yu et al. [5] formed microstructures on glass by using a plasma etching method to generate a superhydrophilic surface which mimics the self-cleaning of lotus leaf. Although superhydrophilic surfaces were widely exploited, Liquid Infused Surfaces(LISs) have attracted much attention from scientists recently because of their special surface wetting properties [6,7]. Comparing to other strategies, LISs have advantages due to its availability, relatively low cost, high processing speed, and wide variety of patterning effects. For instance, Sunny et al. [4] reported an anti-fouling material fabricated via the layer-by-layer (LbL) deposition of charged particles to form hierarchical micro porous structures with infiltrated lubricants.”

  1. Point 12: The author shows a schematic of PS/SBS porous film formation in Fig. 1. However, there are some misunderstandings between the panels and the description. It is difficult to relate a panel to a fabrication step, especially panels after the spin-coating panel.

Response 12: Thanks for your kind comment. The Figure1 have been modified with more expression as shown in corresponding position of the paper. “After coating, the glass substrate with solution was dried at room temperature for 3-5 minutes to obtain a uniformly and orderly porous film structure (Figure 1e). Subsequently, 20μL cm2 silicone oil was added onto the glass substrate until spread over the entire surface. Finally, the substrates were positioned vertically for 3 h to drain off the excess silicone oil (Figure 1f-g).”

  1. Point 13: For figure 1, the author mentioned that PS:SBS = 3:1. However, from the panel, the ratio seems to be 1:1. And the panel after spin-coating panel and the middle panel are hard for readers to understand.

Response 13: Thanks for your kind comment. The Figure1 have been modified with more expression as shown in corresponding position of the paper.

  1. Point 14: To confirm the structure of the holes, please provide some SEM with higher magnification to clearly observe the edge of the pore. And side-view SEM is also needed.

Response 14: Thanks for your kind comment. The SEM photo is added in Figure 2.The side-view SEM photo is added as Figure 3 in Supporting Information.

  1. Point 15: The detail of the experiment should be described in Methods, such as how to deal with the porous film with lubricant, how to measure the film thickness, contact angle.

Response 15: Thanks for your kind comment. The relevant descriptions have been modified with more expression as shown in Page 3, line 101. “20μL cm2 silicone oil was added onto the glass substrate until spread over the entire surface. Finally, the substrates were positioned vertically for 3 h to drain off the excess silicone oil” “Contact angles (CAs) and sliding angles (SAs) were measured by use of the droplet volume of  5μL. The transmittance of the samples over the entire visible spectrum, 400–800 nm, was recorded using a spectrophotometer. The film thickness was measured by a profilometer.”

  1. Point 16: How about the liquid-repelling feature of the PS/SBS film without lubricant?

Response 16: Thanks for your good comment. The relevant descriptions have been modified as shown in Page 6, line 202. “The wettabilities of seven different droplets were compared on the glass substrates of porous films between with and without silicone oil ” “As shown in Figure 5a and S4, the wet-slip porous film consistently showed a reduced contact angle comparing to the dry-slip porous film and a low slip angle ≤5°.” The Contact angles of different droplets on the surface of a dry porous membrane is added in Supporting Information Fig 4.

  1. Point 17: Please unify the font in figure 6b.

Response 17: Thanks for your good comment. The Figure6b have been modified as shown in corresponding position of the paper.

  1. Point 18: For figure 4, panel c compares the film with/without silicone oil. But the two birds in the two pictures are not the same.

Response 18: Thanks for your good comment. The Figure4 have been modified as shown in corresponding position of the paper.

  1. Point 19: It is better to briefly summarize the conclusion of the previous work than solely saying “the optimal ones” (line 91)

Response 19: Thanks for your good comment. The relevant descriptions have been modified as shown in Page 5, line 182. “dry porous film and super wet slip porous film(dealt with silicone oil) were compared under the parameters of concentration of 40g/L, spin-coating speed of 300r/min and spin-coating time of 6s which was analyzed as the optimum in the previous section.”

  1. Point 20: The figure description of Figure 1 and Figure 2 are too general. More content, such as why PS and PB are connected and what the central picture represents, should be added.

Response 20: Thanks for your good comment. The Figure1 and Figure 2 have been modified with more expression as shown in corresponding position of the paper. “Figure 1. Schematic diagram of PS/SBS porous film formation. (a) PS and SBS mix in a beaker at the ratio of 3:1. (b) CH2Cl2 is added in. (c) PS and SBS are completely dissolved. (d) the prepared solution is dipped onto a circular curved glass substrate and spun. (e) the glass substrate with solution is dried. (f) silicone oil is added onto the glass substrate. (g) the orderly porous film is obtained on curved surface. (h) the SEM photo of porous film.” “Figure 2. The process of the porous films formation mechanism.(a) PS/SBS solution remain stabilized on the glass and solvent evaporate. (b) The dichloromethane continues to evaporate. (c) The PS interactions lead to the formation of holes. (d) The porous structure is formed after solvent evaporate completely.”

  1. Point 21: The author also misses the latest articles in their citation.  Rev.2022, 122, 5, 5233–5276. Trends in Chemistry, Volume 2, Issue 6, June 2020, Pages 519-534. Matter, Volume 2, Issue 4, 1 April 2020, Pages 948-964.

Response 21: Thanks for your good comment. The relevant citation have been modified in corresponding position of the paper.

Reviewer 2 Report

This work by Yang Liu and co-authors reports an excellent liquid-infused surface. In this work, the authors focus on synthesizing porous PS/ SBS films by a straightforward method. It is achieved by spin-coating on a glass substrate with PS/SBS solution applied dropwise. However, the author should address my concern before they get final published.

1. For the introduction part, the logic between paragraphs is not clearly shown. And at the end of the second paragraph of the introduction, there is a problem with the placement of punctuation and citations.

2. Introduction should also mention some other strategies beside liquid-infused surfaces (LISs) and describe the advantages of LISs over others. Besides, the introduction also needs to include at least one prototypical work of LISs with more details.

3. The author shows a schematic of PS/SBS porous film formation in Fig. 1. However, there are some misunderstandings between the panels and the description. It is difficult to relate a panel to a fabrication step, especially panels after the spin-coating panel.

4. For figure 1, the author mentioned that PS:SBS = 3:1. However, from the panel, the ratio seems to be 1:1. And the panel after spin-coating panel and the middle panel are hard for readers to understand.

5. To confirm the structure of the holes, please provide some SEM with higher magnification to clearly observe the edge of the pore. And side-view SEM is also needed.

6. The detail of the experiment should be described in Methods, such as how to deal with the porous film with lubricant, how to measure the film thickness, contact angle.

7. How about the liquid-repelling feature of the PS/SBS film without lubricant?

8. Please unify the font in figure 6b.

9. For figure 4, panel c compares the film with/without silicone oil. But the two birds in the two pictures are not the same.

10. It is better to briefly summarize the conclusion of the previous work than solely saying “the optimal ones” (line 91)

11. The figure description of Figure 1 and Figure 2 are too general. More content, such as why PS and PB are connected and what the central picture represents, should be added.

12. The author also misses the latest articles in their citation. Chem. Rev. 2022, 122, 5, 5233–5276. Trends in Chemistry, Volume 2, Issue 6, June 2020, Pages 519-534. Matter, Volume 2, Issue 4, 1 April 2020, Pages 948-964.

Author Response

(The authors gave the same response as above.)

Round 2

Reviewer 2 Report

I agree to publish this article to Micromachines.